# ON-POLICY TRUST REGION POLICY OPTIMISATION WITH REPLAY BUFFERS

## ABSTRACT

Building upon the recent success of deep reinforcement learning methods, we investigate the possibility of on-policy reinforcement learning improvement by reusing the data from several consecutive policies. On-policy methods bring many benefits, such as ability to evaluate each resulting policy. However, they usually discard all the information about the policies which existed before. In this work, we propose adaptation of the replay buffer concept, borrowed from the off-policy learning setting, to create the method, combining advantages of on- and off-policy learning. To achieve this, the proposed algorithm generalises the $Q$-, value and advantage functions for data from multiple policies. The method uses trust region optimisation, while avoiding some of the common problems of the algorithms such as TRPO or ACKTR: it uses hyperparameters to replace the trust region selection heuristics, as well as the trainable covariance matrix instead of the fixed one. In many cases, the method not only improves the results comparing to the state-of-the-art trust region on-policy learning algorithms such as PPO, ACKTR and TRPO, but also with respect to their off-policy counterpart DDPG.

## 1 INTRODUCTION

The past few years have been marked by active development of reinforcement learning methods. Although the mathematical foundations of reinforcement learning have been known long before (Sutton & Barto, 1998), starting from 2013, the novel deep learning techniques allowed to solve vision based discrete control tasks such as Atari 2600 games (Mnih et al., 2013) as well as continuous control problems (Lillicrap et al., 2015; Mnih et al., 2016). Many of the leading state-of-the-art reinforcement learning methods share the *actor-critic* architecture (Crites & Barto, 1995). Actor-critic methods separate the actor, providing a policy, and the critic, providing an approximation for the expected discounted cumulative reward or some derived quantities such as advantage functions (Baird III, 1993). However, despite improvements, state-of-the-art reinforcement learning still suffers from poor sample efficiency and extensive parameterisation. For most real-world applications, in contrast to simulations, there is a need to learn in real time and over a limited training period, while minimising any risk that would cause damage to the actor or the environment.

Reinforcement learning algorithms can be divided into two groups: *on-policy* and *off-policy* learning. On-policy approaches (e. g., SARSA (Rummery & Niranjan, 1994), ACKTR (Wu et al., 2017)) evaluate the target policy by assuming that future actions will be chosen according to it, hence the exploration strategy must be incorporated as a part of the policy. Off-policy methods (e. g., Q-learning (Watkins, 1989), DDPG (Lillicrap et al., 2015)) separate the exploration strategy, which modifies the policy to explore different states, from the target policy.

The off-policy methods commonly use the concept of replay buffers to memorise the outcomes of the previous policies and therefore exploit the information accumulated through the previous iterations (Lin, 1993). Mnih et al. (2013) combined this *experience replay* mechanism with Deep Q-Networks (DQN), demonstrating end-to-end learning on Atari 2600 games. One limitation of DQN is that it can only operate on discrete action spaces. Lillicrap et al. (2015) proposed an extension of DQN to handle continuous action spaces based on the Deep Deterministic Policy Gradient (DDPG). There, exponential smoothing of the target actor and critic weights has been introduced to ensure stability of the rewards and critic predictions over the subsequent iterations. In order to improve the variance of policy gradients, Schulman et al. (2015b) proposed a Generalised Advantage Function. Mnih

et al. (2016) combined this advantage function learning with a parallelisation of exploration using differently trained actors in their Asynchronous Advantage Actor Critic model (A3C); however, Wang et al. (2016) demonstrated that such parallelisation may also have negative impact on sample efficiency. Although some work has been performed on improvement of exploratory strategies for reinforcement learning (Hester et al., 2013), but it still does not solve the fundamental restriction of inability to evaluate the actual policy, neither it removes the necessity to provide a separate exploratory strategy as a separate part of the method.

In contrast to those, state-of-the-art on-policy methods have many attractive properties: they are able to evaluate exactly the resulting policy with no need to provide a separate exploration strategy. However, they suffer from poor sample efficiency, to a larger extent than off-policy reinforcement learning. TRPO method (Schulman et al., 2015a) has introduced trust region policy optimisation to explicitly control the speed of policy evolution of Gaussian policies over time, expressed in a form of Kullback-Leibler divergence, during the training process. Nevertheless, the original TRPO method suffered from poor sample efficiency in comparison to off-policy methods such as DDPG. One way to solve this issue is by replacing the first order gradient descent methods, standard for deep learning, with second order natural gradient (Amari, 1998). Wu et al. (2017) used a Kronecker-factored Approximate Curvature (K-FAC) optimiser (Martens & Grosse, 2015) in their ACKTR method. PPO method (Schulman et al., 2017) proposes a number of modifications to the TRPO scheme, including changing the objective function formulation and clipping the gradients. Wang et al. (2016) proposed another approach in their ACER algorithm: in this method, the target network is still maintained in the off-policy way, similar to DDPG (Lillicrap et al., 2015), while the trust region constraint is built upon the difference between the current and the target network.

Related to our approach, recently a group of methods has appeared in an attempt to get the benefits of both groups of methods. Gu et al. (2017) propose interpolated policy gradient, which uses the weighted sum of both stochastic (Sutton et al., 2000) and deterministic policy gradient (Silver et al., 2014). Nachum et al. (2018) propose an off-policy trust region method, Trust-PCL, which exploits off-policy data within the trust regions optimisation framework, while maintaining stability of optimisation by using relative entropy regularisation.

While it is a common practice to use replay buffers for the off-policy reinforcement learning, their existing concept is not used in combination with the existing on-policy scenarios, which results in discarding all policies but the last. Furthermore, many on-policy methods, such as TRPO (Schulman et al., 2015a), rely on stochastic policy gradient (Sutton et al., 2000), which is restricted by stationarity assumptions, in a contrast to those based on deterministic policy gradient (Silver et al., 2014), like DDPG (Lillicrap et al., 2015). In this article, we describe a novel reinforcement learning algorithm, allowing the joint use of replay buffers with trust region optimisation and leading to sample efficiency improvement. The contributions of the paper are given as follows:

1. a reinforcement learning method, enabling replay buffer concept along with on-policy data;

2. theoretical insights into the replay buffer usage within the on-policy setting are discussed;

3. we show that, unlike the state-of-the-art methods as ACKTR (Wu et al., 2017), PPO (Schulman et al., 2017) and TRPO (Schulman et al., 2015a), a single non-adaptive set of hyperparameters such as the trust region radius is sufficient for achieving better performance on a number of reinforcement learning tasks.

As we are committed to make sure the experiments in our paper are repeatable and to further ensure their acceptance by the community, we will release our source code shortly after the publication.

## 2 BACKGROUND

### 2.1 ACTOR-CRITIC REINFORCEMENT LEARNING

Consider an agent, interacting with the environment by responding to the states $s_t$, $t \geq 0$, from the state space $\mathcal{S}$, which are assumed to be also the observations, with actions $a_t$ from the action space $\mathcal{A}$ chosen by the policy distribution $\pi_\theta(\cdot|s_t)$, where $\theta$ are the parameters of the policy. The initial state distribution is $\rho_0 : \mathcal{S} \rightarrow \mathbb{R}$. Every time the agent produces an action, the environment gives back a reward $r(s_t, a_t) \in \mathbb{R}$, which serves as a feedback on how good the action choice was and

switches to the next state $s_{t+1}$ according to the transitional probability $P(s_{t+1}|s_t, a_t)$. Altogether, it can be formalised as an infinite horizon $\gamma$-discounted Markov Decision Process $(\mathcal{S}, \mathcal{A}, P, r, \rho_0, \gamma)$, $\gamma \in [0, 1)$ (Wu et al., 2017; Schulman et al., 2015a). The expected discounted return (Bellman, 1957) is defined as per Schulman et al. (2015a):

$$\rho(\pi) = \mathbb{E}_{s_0, a_0, \cdots} \left[ \sum_{t=0}^{\infty} \gamma^t r(s_t, a_t) \right] \tag{1}$$

The advantage function $A^\pi$ (Baird III, 1993), the value function $V^\pi$ and the $Q$-function $Q^\pi$ are defined as per Mnih et al. (2016); Schulman et al. (2015a):

$$A^\pi(s, a) = Q^\pi(s, a) - V^\pi(s), \tag{2}$$

$$Q^\pi(s_t, a_t) = \mathbb{E}_{s_{t+1}, a_{t+1}, \ldots} \left[ \sum_{l=0}^{\infty} \gamma^l r(s_{t+l}, a_{t+l}) \right], t \geq 0, \tag{3}$$

$$V^\pi(s_t) = \mathbb{E}_{a_t, s_{t+1}, \ldots} \left[ \sum_{l=0}^{\infty} \gamma^l r(s_{t+l}, a_{t+l}) \right], t \geq 0 \tag{4}$$

In all above definitions $s_0 \sim \rho_0(s_0)$, $a_t \sim \pi(a_t|s_t)$, $s_{t+1} \sim P(s_{t+1}|s_t, a_t)$, and the policy $\pi = \pi_\theta$ is defined by its parameters $\theta$.

## 2.2 TRUST REGION POLICY OPTIMISATION (TRPO)

A straightforward approach for learning a policy is to perform unconstrained maximisation $\rho(\pi_\theta)$ with respect to the policy parameters $\theta$. However, for the state-of-the-art iterative gradient-based optimisation methods, this approach would lead to unpredictable and uncontrolled changes in the policy, which would impede efficient exploration. Furthermore, in practice the exact values of $\rho(\pi_\theta)$ are unknown, and the quality of its estimates depends on approximators which tend to be correct only in the vicinity of parameters of observed policies.

Schulman et al. (2015a), based on theorems by Kakade (2002), prove the minorisation-maximisation (MM) algorithm (Hunter & Lange, 2004) for policy parameters optimisation. Schulman et al. (2015a) mention that in practice the algorithm's convergence rate and the complexity of maximum KL divergence computations makes it impractical to apply this method directly. Therefore, they proposed to replace the unconstrained optimisation with a similar constrained optimisation problem, the Trust Region Policy Optimisation (TRPO) problem:

$$\arg\max_\theta \rho(\pi_\theta) \tag{5}$$

$$D_{KL}(\pi_{\theta_{\text{old}}}, \pi_\theta) \leq \delta, \tag{6}$$

where $D_{KL}$ is the KL divergence between the old and the new policy $\pi_{\theta_{\text{old}}}$ and $\pi_\theta$ respectively, and $\delta$ is the trust region radius. Despite this improvement, it needs some further enhancements to solve this problem efficiently, as we will elaborate in the next section.

## 2.3 SECOND ORDER ACTOR-CRITIC NATURAL GRADIENT OPTIMISATION

Many of the state-of-the-art trust region based methods, including TRPO (Schulman et al., 2015a) and ACKTR (Wu et al., 2017), use second order natural gradient based actor-critic optimisation (Amari, 1998; Kakade, 2002). The motivation behind it is to eliminate the issue that gradient descent loss, calculated as the Euclidean norm, is dependent on parametrisation. For this purpose, the Fisher information matrix is used, which is, as it follows from Amari (1998) and Kakade (2002), normalises per-parameter changes in the objective function. In the context of actor-critic optimisation it can be written as (Wu et al., 2017; Kakade, 2002), where $p(\tau)$ is the trajectory distribution $p(s_0) \prod_{t=0}^{T} \pi(a_t|s_t) p(s_{t+1}|s_t, a_t)$:

$$F = \mathbb{E}_{p(\tau)} \left[ \nabla_\theta \log \pi(a_t|s_t) \left( \nabla_\theta \log \pi(a_t|s_t) \right)^T \right]. \tag{7}$$

However, the computation of the Fisher matrix is intractable in practice due to the large number of parameters involved; therefore, there is a need to resort to approximations, such as the Kronecker-factored approximate curvature (K-FAC) method (Martens & Grosse, 2015), which has been first proposed for ACKTR in (Wu et al., 2017). In the proposed method, as it is detailed in Algorithm 1, this optimisation method is used for optimising the policy.

## 3 METHOD DESCRIPTION

While the original trust regions optimisation method can only use the samples from the very last policy, discarding the potentially useful information from the previous ones, we make use of samples over several consecutive policies. The rest of the section contains definition of the proposed replay buffer concept adaptation, and then formulation and discussion of the proposed algorithm.

### 3.1 USAGE OF REPLAY BUFFERS

Mnih et al. (2013) suggested to use replay buffers for DQN to improve stability of learning, which then has been extended to other off-policy methods such as DDPG (Lillicrap et al., 2015). The concept has not been applied to on-policy methods like TRPO (Schulman et al., 2015a) or ACKTR (Wu et al., 2017), which do not use of previous data generated by other policies. Although based on trust regions optimisation, ACER (Wang et al., 2016) uses replay buffers for its off-policy part.

In this paper, we propose a different concept of the replay buffers, which combines the on-policy data with data from several previous policies, to avoid the restrictions of policy distribution stationarity for stochastic policy gradient (Sutton et al., 2000). Such replay buffers are used for storing simulations from several policies at the same time, which are then utilised in the method, built upon generalised value and advantage functions, accommodating data from these policies. The following definitions are necessary for the formalisation of the proposed algorithm and theorems.

We define a generalised $Q$-function for multiple policies $\{\pi_1, \ldots, \pi_n, \ldots, \pi_N\}$ as

$$Q^{\overline{\pi}}(s_t, a_t) = \mathbb{E}_n \mathbb{E}_{s_{t+1}^n, a_{t+1}^n, \ldots} \left[ \sum_{l=0}^{\infty} \gamma^l r(s_{t+l}^n, a_{t+l}^n) \right], t \geq 0, \tag{8}$$

$$s_0^n \sim \rho_0(s_0^n), s_{t+1}^n \sim P(s_{t+1}^n | s_t^n, a_t^n), a_t^n \sim \pi_n(a_t^n | s_t^n). \tag{9}$$

We also define the generalised value function and the generalised advantage function as

$$V^{\overline{\pi}}(s_t) = \mathbb{E}_n \mathbb{E}_{a_t, s_{t+1}, a_{t+1}} \left[ \sum_{l=0}^{\infty} \gamma^l r(s_{t+l}^n, a_{t+l}^n) \right], t \geq 0, \tag{10}$$

$$A^{\pi_n}(s_t) = Q^{\pi_n}(s_t, a_t) - V^{\overline{\pi}}(s_t), t \geq 0, \tag{11}$$

To conform with the notation from Sutton et al. (2000), we define

$$\rho(\overline{\pi}) = V^{\overline{\pi}}(s_0), D^{\pi_n}(s) = \sum_{k=0}^{\infty} \gamma^k P(s_0 \to s, k, \pi_n), \tag{12}$$

$P(s \to x, k, \pi)$, as in Sutton et al. (2000), is the probability of transition from the state $s$ to the state $x$ in $k$ steps using policy $\pi$.

**Theorem 1.** *For the set of policies $\{\pi_1, \ldots, \pi_N\}$ the following equality will be true for the gradient:*

$$\frac{\partial \rho^{\overline{\pi}}}{\partial \theta} = \sum_{n=1}^{N} p(\pi_n) \int_s ds D^{\pi_n}(s) \int_a da \frac{\partial \pi_n(s, a)}{\partial \theta} [Q^{\pi_n}(s, a) + b^{\pi_n}(s)], \tag{13}$$

*where $\theta$ are the joint parameters of all policies $\{\pi_n\}$ and $b^{\pi_n}(s)$ is a bias function for the policy.*

The proof of Theorem 1 is given in Appendix B. Applying a particular case of the bias function $b^{\pi_n}(s) = -V^{\overline{\pi}}(s)$ and using the likelihood ratio transformation, one can get

$$\frac{\partial \rho^{\overline{\pi}}}{\partial \theta} = \sum_{n=1}^{N} p(\pi_n) \int_s ds D^{\pi_n}(s) \int_a da \pi_n(s, a) \frac{\partial \log \pi_n(s, a)}{\partial \theta} A^{\pi_n}(s, a) \tag{14}$$

### 3.2 ALGORITHM DESCRIPTION

The proposed approach is summarised in Algorithm 1. The replay buffer $R_p$ contains data collected from several subsequent policies. The size of this buffer is RBP_CAPACITY.

---

**Algorithm 1** Trust Regions Algorithm with a Replay Buffer

---

{**Initialisation**} Randomly initialise the weights $\theta$ of the policy estimator $\pi(\cdot)$ and $\Psi$ of the value function estimator $\tilde{V}(\cdot)$. Initialise the policy replay buffer $R_p = \{\}$, set $i = 0, \delta = \text{DELTA}$
**while** $i < \text{MAX\_TIMESTEPS}$ **do**
    {**Stage 1**} Collect $n^i$ data paths using the current policy $\pi(\cdot)$: $P = \left\{ \left\langle (s_0^j, a_0^j, r_0^j), \ldots, (s_{k_j}^j, a_{k_j}^j, r_{k_j}^j) \right\rangle \right\}_{j=0}^{n^i}$, increase $i$ by the total number of timesteps in all new paths.
    {**Stage 2**} Put recorded paths into the policy paths replay buffer $R_p \leftarrow P$.
    {**Stage 3**} For every path in $R_p$ compute the targets for the value function regression using equation (15). $\Psi$ = Update the value function estimator parameters
    {**Stage 4**} For every path in $R_p$, estimate the advantage function using Equation (23).
    {**Stage 5**} Update parameters of the policy $\theta$ for N\_ITER\_PL\_UPDATE iterations using the gradient from Equation (25) and a barrier function defined in Equation (26).
**end while**

---

During Stage 1, the data are collected for every path until the termination state is received, but at least TIMESTEPS\_PER\_BATCH steps in total for all paths. The policy actions are assumed to be sampled from the Gaussian distribution, with the mean values predicted by the policy estimator along with the covariance matrix diagonal. The covariance matrix output was inspired, although the idea is different, by the EPG paper (Ciosek & Whiteson, 2017).

At Stage 2, the obtained data for every policy are saved in the policy replay buffer $R_p$.

At Stage 3, the regression of the value function is trained using Adam optimiser (Kingma & Ba, 2015) with step size VF\_STEP\_SIZE for N\_ITER\_VF\_UPDATE iterations. For this regression, the sum-of-squares loss function is used. The value function target values are computed for every state $s_t$ for every policy in the replay buffer using the actual sampled policy values, where $t_{\max}$ is the maximum policy step index:

$$\hat{V}(s_t) = \sum_{l=0}^{t_{\max}-t} \gamma^l r(s_{t+l}, a_{t+l}), \tag{15}$$

During Stage 4, we perform the advantage function estimation.

Schulman et al. (2015b) proposed the Generalised Advantage Estimator for the advantage function $A^\pi(s_t, a_t)$ as follows:

$$\tilde{A}^\pi(s_t, a_t) = (1 - \lambda)(\hat{A}_t^{\pi,(1)} + \lambda \hat{A}_t^{\pi,(2)} + \lambda^2 \hat{A}_t^{\pi,(3)} + \ldots), \tag{16}$$

where

$$\hat{A}_t^{\pi,(1)} = -\tilde{V}^\pi(s_t) + r_t + \gamma \tilde{V}^\pi(s_{t+1}), \ldots \tag{17}$$

$$\hat{A}_t^{\pi,(k)} = -\tilde{V}^\pi(s_t) + r_t + \ldots + \gamma^{k-1} r_{t+k-1} + \gamma^k \tilde{V}^\pi(s_{t+k}), \ldots \tag{18}$$

Here $k > 0$ is a cut-off value, defined by the length of the sequence of occured states and actions within the MDP, $\lambda \in [0, 1]$ is an estimator parameter, and $\tilde{V}^\pi(s_t)$ is the approximation for the value function $V_\pi(s_t)$, with the approximation targets defined in Equation (15). As proved in Schulman et al. (2015b), after rearrangement this would result in the generalised advantage function estimator

$$\tilde{A}^\pi(s_t, a_t) = \sum_{l=0}^{k-1} (\gamma\lambda)^l (r_{t+l} + \gamma \tilde{V}^\pi(s_{t+l+1}) - \tilde{V}^\pi(s_{t+l})), \tag{19}$$

For the proposed advantage function (see Equation 11), the estimator could be defined similarly to Schulman et al. (2015b) as

$$\tilde{A}^{\pi_n}(s_t, a_t) = (1 - \lambda)(\hat{A}_t^{\pi_n,(1)} + \lambda \hat{A}_t^{\pi_n,(2)} + \lambda^2 \hat{A}_t^{\pi_n,(3)} + \ldots), \tag{20}$$

$$\hat{A}_t^{\pi_n,(1)} = -\tilde{V}^{\overline{\pi}}(s_t) + r_t + \gamma \tilde{V}^{\pi_n}(s_{t+1}), \ldots \tag{21}$$

$$\hat{A}_t^{\pi_n,(k)} = -\tilde{V}^{\overline{\pi}}(s_t) + r_t + \gamma r_{t+1} + \ldots + \gamma^{k-1} r_{t+k-1} + \gamma^k \tilde{V}^{\pi_n}(s_{t+k}). \tag{22}$$

However, it would mean the estimation of multiple value functions, which diminishes the replay buffer idea. To avoid it, we modify this estimator for the proposed advantage function as

$$\tilde{A}^{\pi_n}(s_t, a_t) = \sum_{l=0}^{k-1} (\gamma\lambda)^l (r_{t+l} + \gamma\tilde{V}^{\overline{\pi}}(s_{t+l+1}) - \tilde{V}^{\overline{\pi}}(s_{t+l})), \tag{23}$$

**Theorem 2.** *The difference between the estimators (20) and (23) is*

$$\Delta\tilde{A}^{\pi_n}(s_t, a_t) = \gamma(1-\lambda) \sum_{l=1}^{k} \lambda^{l-1}\gamma^{l-1}(\tilde{V}^{\pi_n}(s_{t+l}) - \tilde{V}^{\overline{\pi}}(s_{t+l})). \tag{24}$$

The proof of Theorem 2 is given in Appendix C. It shows that the difference between two estimators is dependent of the difference in the conventional and the generalised value functions; given the continuous value function approximator it reveals that the closer are the policies, within a few trust regions radii, the smaller will be the bias.

During Stage 5, the policy function is approximated, using the K-FAC optimiser (Martens & Grosse, 2015) with the constant step size PL_STEP_SIZE. As one can see from the description, and differently from ACKTR, we do not use any adaptation of the trust region radius and/or optimisation algorithm parameters. Also, the output parameters include the diagonal of the (diagonal) policy covariance matrix. The elements of the covariance matrix, for the purpose of efficient optimisation, are restricted to universal minimum and maximum values MIN_COV_EL and MAX_COV_EL.

As an extention from Schulman et al. (2015b) and following Theorem 1 with the substitution of likelihood ratio, the policy gradient estimation is defined as

$$\nabla\rho(\theta) \approx \mathbb{E}_n \mathbb{E}_{\pi_n} \left[ \sum_{t=0}^{\infty} \tilde{A}^{\pi_n}(s_t^n, a_t^n) \nabla_\theta \log \pi_n(a_t^n | s_t^n) \right]. \tag{25}$$

To practically implement this gradient, we substitute the parameters $\theta^\pi$, derived from the latest policy for the replay buffer, instead of joint $\theta$ parameters assuming that the parameters would not deviate far from each other due to the trust region restrictions; it is still possible to calculate the estimation of $\tilde{A}^{\pi_n}(s_t^n, a_t^n)$ for each policy using Equation (23) as these policies are observed. For the constrained optimisation we add the linear barrier function to the function $\rho(\theta)$:

$$\rho_b(\theta) = \rho(\theta) - \alpha \cdot \max(0, D_{KL}(\pi_{\theta_{\text{old}}}, \pi_\theta) - \delta), \tag{26}$$

where $\alpha > 0$ is a barrier function parameter and $\theta_{\text{old}}$ are the parameters of the policy on the previous iteration. Besides of removing the necessity of heuristical estimation of the optimisation parameters, it also conforms with the theoretical prepositions shown in Schulman et al. (2017) and, while our approach is proposed independently, pursues the similar ideas of using actual constrained optimisation method instead of changing the gradient step size parameters as per Schulman et al. (2015a).

The networks' architectures correspond to OpenAI Baselines ACKTR implementation (Dhariwal et al., 2017) ,which has been implemented by the ACKTR authors (Wu et al., 2017). The only departure from the proposed architecture is the diagonal covariance matrix outputs, which are present, in addition to the mean output, in the policy network.

## 4 EXPERIMENTS

### 4.1 EXPERIMENTAL RESULTS

In order to provide the experimental evidence for the method, we have compared it with the on-policy ACKTR (Wu et al., 2017), PPO (Schulman et al., 2017) and TRPO (Schulman et al., 2015a) methods, as well as with the off-policy DDPG (Lillicrap et al., 2015) method on the MuJoCo (Todorov et al., 2012) robotic simulations. The technical implementation is described in Appendix A.

Figure 1 shows the total reward values and their standard deviations, averaged over every one hundred simulation steps over three randomised runs. The results show drastic improvements over the state-of-the-art methods, including the on-policy ones (ACKTR, TRPO, PPO), on most problems.

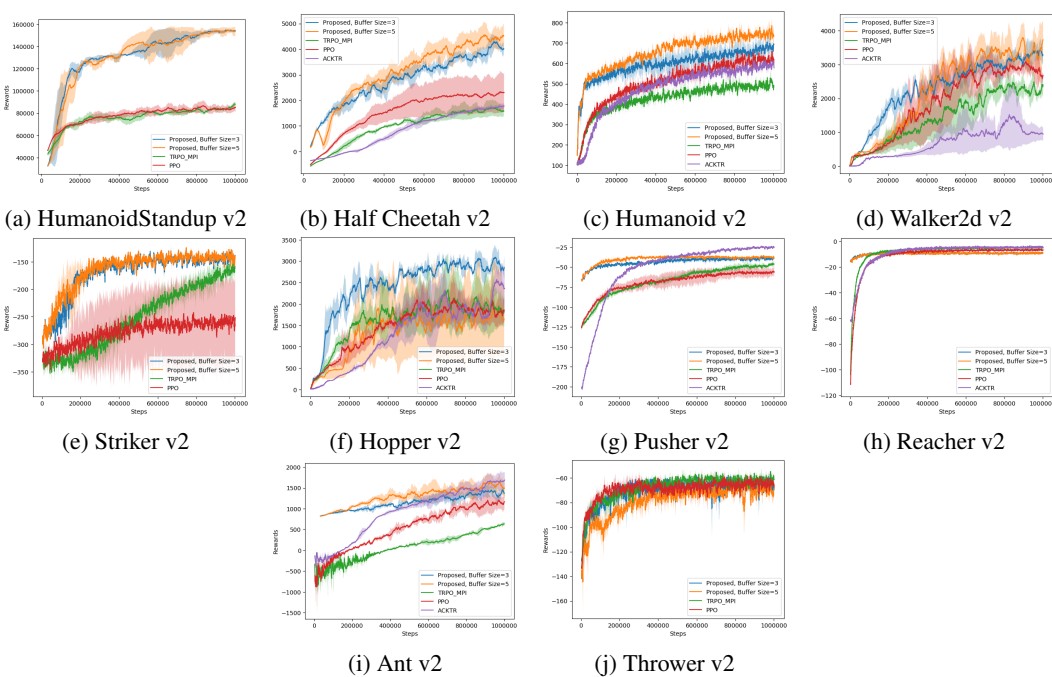

Figure 1: Experiments in MuJoCo environment (Todorov et al., 2012): comparison with TRPO (Schulman et al., 2015a), ACKTR (Wu et al., 2017) and PPO (Schulman et al., 2017)

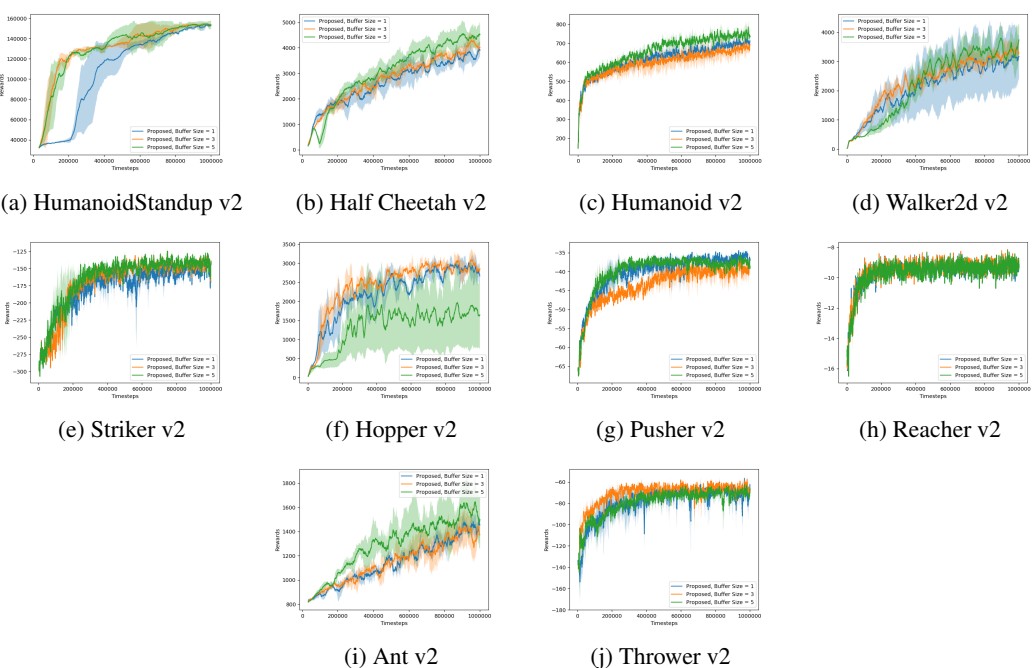

Figure 2: Experiments in MuJoCo environment (Todorov et al., 2012): comparison between different replay buffer depth values

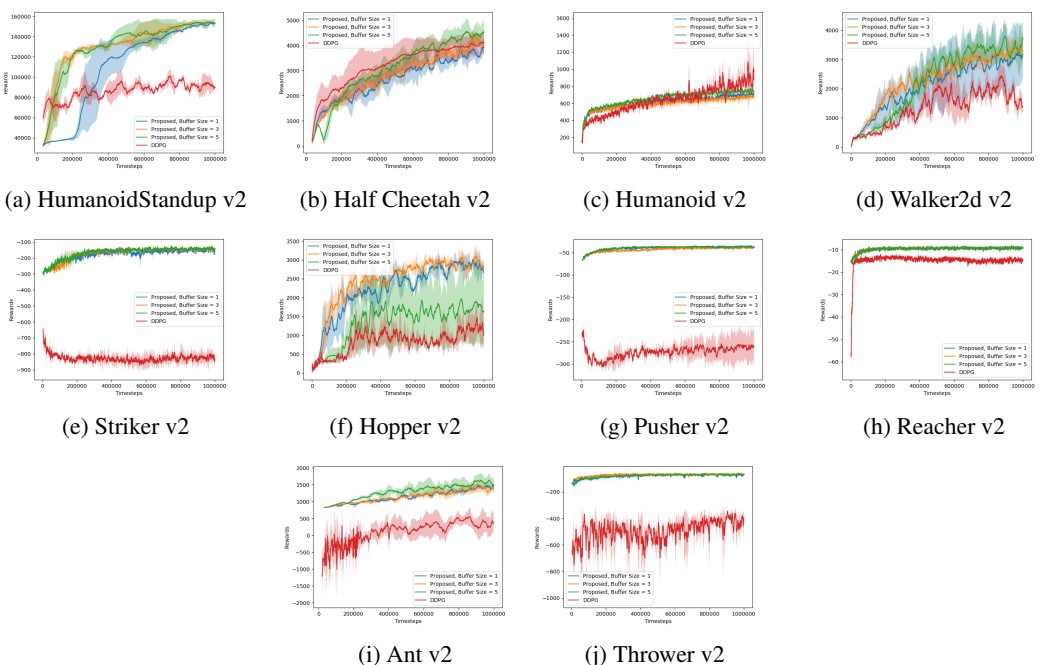

(a) HumanoidStandup v2    (b) Half Cheetah v2    (c) Humanoid v2    (d) Walker2d v2

(e) Striker v2    (f) Hopper v2    (g) Pusher v2    (h) Reacher v2

(i) Ant v2    (j) Thrower v2

Figure 3: Experiments in MuJoCo environment (Todorov et al., 2012): comparison between the proposed algorithm and DDPG (Lillicrap et al., 2015)

In contrast to those methods, the method shows that the adaptive values for trust region radius can be advantageously replaced by a fixed value in a combination with the trainable policy distribution covariance matrix, thus reducing the number of necessary hyperparameters. The results for ACKTR for the tasks HumanoidStandup, Striker and Thrower are not included as the baseline ACKTR implementation (Dhariwal et al., 2017) diverged at the first iterations with the predefined parameterisation. PPO results are obtained from baselines implementation PPO1 (Dhariwal et al., 2017).

Figure 2 compares results for different replay buffer sizes; the size of the replay buffers reflects the number of policies in it and not actions (i.e. buffer size 3 means data from three successive policies in the replay buffer). We see that in most of the cases, the use of replay buffers show performance improvement against those with replay buffer size 1 (i.e., no replay buffer with only the current policy used for policy gradient); substantial improvements can be seen for HumanoidStandup task.

Figure 3 shows the performance comparison with the DDPG method (Lillicrap et al., 2015). In all the tasks except HalfCheetah and Humanoid, the proposed method outperforms DDPG. For HalfChee-tah, the versions with a replay buffer marginally overcomes the one without. It is also remarkable that the method demonstrates stable performance on the tasks HumanoidStandup, Pusher, Striker and Thrower, on which DDPG failed (and these tasks were not included into the DDPG article).

## 5    CONCLUSION

The paper combines replay buffers and on-policy data for reinforcement learning. Experimental results on various tasks from the MuJoCo suite (Todorov et al., 2012) show significant improvements compared to the state of the art. Moreover, we proposed a replacement of the heuristically calculated trust region parameters, to a single fixed hyperparameter, which also reduces the computational expences, and a trainable diagonal covariance matrix.

The proposed approach opens the door to using a combination of replay buffers and trust regions for reinforcement learning problems. While it is formulated for continuous tasks, it is possible to reuse the same ideas for discrete reinforcement learning tasks, such as ATARI games.

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

## A  TECHNICAL IMPLEMENTATION

The parameters of Algorithm 1, used in the experiment, are given in Table 1; the parameters were initially set, where possible, to the ones taken from the state-of-the-art trust region approach implementation (Wu et al., 2017; Dhariwal et al., 2017), and then some of them have been changed based on the experimental evidence. As the underlying numerical optimisation algorithms are out of the scope of the paper, the parameters of K-FAC optimiser from Dhariwal et al. (2017) have been used for the experiments; for the Adam algorithm (Kingma & Ba, 2015), the default parameters from Tensorflow (Abadi et al., 2016) implementation ($\beta_1 = 0.9, \beta_2 = 0.999, \epsilon = 1 \cdot 10^{-8}$) have been used.

The method has been implemented in Python 3 using Tensorflow (Abadi et al., 2016) as an extension of the OpenAI baselines package (Dhariwal et al., 2017). The neural network for the control experiments consists of two fully connected layers, containing 64 neurons each, following the OpenAI ACKTR network implementation (Dhariwal et al., 2017).

| Parameter | Value |
|---|---|
| $\alpha$ | 100 |
| TIMESTEPS_PER_BATCH | 1000 |
| VF_STEP_SIZE | $1 \cdot 10^{-3}$ |
| PL_STEP_SIZE | $1 \cdot 10^{-2}$ |
| DELTA | 0.1 |
| N_ITER_VF_UPDATE | 100 |

| Parameter | Value |
|---|---|
| N_ITER_PL_UPDATE | 10 |
| RBP_CAPACITY | 3 |
| MAX_TIMESTEPS | 1000000 |
| MIN_COV_EL | 0.2 |
| MAX_COV_EL | 5.0 |
| $\gamma$ | 0.99 |
| $\lambda$ | 0.97 |

Table 1: The parameters of Algorithm 1

## B  PROOF OF THEOREM 1

*Proof.* Extending the derivation from Sutton et al. (2000), one can see that:

$$\frac{\partial V^{\pi}(s)}{\partial \theta} \stackrel{\text{def}}{=} \frac{\partial}{\partial \theta} \int_a da \pi(s,a)(Q^{\pi}(s,a) + b^{\pi}(s)) =$$

$$\int_x dx \sum_{k=0}^{\infty} \gamma^k P(s \to x, k, \pi) \int_a da \frac{\partial \pi(x,a)}{\partial \theta}(Q^{\pi}(x,a) + b^{\pi}(x)) \quad (27)$$

Then,

$$\frac{\partial \rho^{\overline{\pi}}}{\partial \theta} = \sum_{n=1}^{N} p(\pi_n) \frac{\partial V^{\pi_n}(s_0)}{\partial \theta} =$$

$$\sum_{n=1}^{N} p(\pi_n) \int_s ds \sum_{k=0}^{\infty} \gamma^k P(s_0 \to s, k, \pi_n) \int_a da \frac{\partial \pi_n(s,a)}{\partial \theta}(Q^{\pi_n}(s,a) + b^{\pi_n}(s)) =$$

$$\sum_{n=1}^{N} p(\pi_n) \int_s ds D^{\pi_n}(s) \int_a da \frac{\partial \pi_n(s,a)}{\partial \theta}(Q^{\pi_n}(s,a) + b^{\pi_n}(s)) \quad (28)$$

$\square$

## C  PROOF OF THEOREM 2

*Proof.* The difference between the two $k$-th estimators is given as

$$\Delta \hat{A}_t^{\pi_n,(k)} = \gamma^k (\underbrace{\tilde{V}^{\pi_n}(s_{t+k}) - \tilde{V}^{\overline{\pi}}(s_{t+k})}_{\Delta V^k}) \quad (29)$$

By substituting this into the GAE estimator difference one can obtain

$$\Delta \tilde{A}^{\pi_n}(s_t, a_t) = (1-\lambda)(\gamma \Delta V^1 + \lambda \gamma^2 \Delta V^2 + \lambda^2 \gamma^3 \Delta V^3 + \ldots + \lambda^{k-1} \gamma^k \Delta V^k) =$$

$$\gamma(1-\lambda) \sum_{l=1}^{k} \lambda^{l-1} \gamma^{l-1} \Delta V^l. \quad (30)$$

$\square$

