# OpenReview forum: "On-Policy Trust Region Policy Optimisation with Replay Buffers"
_ICLR.cc/2019/Conference_

### Official Review · AnonReviewer1 · 2018-11-01
**Interesting Generalization of G/V/advantage function but some clarifications are needed.**

**Rating:** 5
**Confidence:** 4

**Review:**

In this paper, the authors present how to integrate replay buffer and on-policy trust region policy optimization (TRPO) by generalizing Q/V/advantage function and then empirically show the proposed method outperforms TRPO/DDPG.

The generalization of advantage function is quite interesting and is well written. One minor issue is that d^{\pi_n} (s) is confusing since it appears after ds.

The theory in Section 3.1 makes sense. However, due to the limitation in Theorem 1 that $\theta$ is the joint parameters, applying Theorem 1 can be difficult. In Eq (25), what is the $\theta$ here? And what does $\nabla_\theta \pi_n$ mean? Does $\pi_n$ uses $\theta$ for computation? One of the problems of using replay buffers in on-policy algorithms is that the stationary distribution of states changes as policy changes, and at least the writing doesn't make it clear on how to solve distribution mismatching issue. Further explanation on Eq (25) might help. If the distributions of states are assumed to match, then the joint distribution of states and actions may mismatch so additional importance sampling might help, as suggested in [1] Eq (3).

Another problem is on the barrier function. In Eq (26), if we only evaluate $\rho_b(\theta)$ (or its gradient w.r.t. $\theta$) at the point $\theta_old$, it doesn't differ with or without the barrier function. So in order to show the barrier function helps, we must evaluate $\rho_b(\theta)$ (or its gradient) at a point $\theta \neq \theta_old$. As far as I know, the underlying optimizer, K-FAC, just evaluates the objective's (i.e., $\rho_b$) gradients at $\theta_old$. Both Conjugate Gradient (CG), which TRPO uses, and K-FAC are trying to solve $F^{-1} g$ where $g$ is the gradient of the objective at the current point.

The experiments show significant improvement over TRPO/DDPG. However, some experiments are also expected.
1. How is the proposed algorithm compared to PPO or Trust PCL?
2. How does the barrier function help? More importantly, what's the comparison of the barrier function to [1] Eq (5)?

The proposed algorithm seems more like a variant of ACKTR instead of TRPO since line search is missing in the proposed algorithm and the underlying optimizer is K-FAC instead of CG.

Ref:
[1]: Proximal Policy Optimization Algorithms, by John Schulman, Filip Wolski, Prafulla Dhariwal, Alec Radford, Oleg Klimov.

---

> ### Author Response · Authors · 2018-11-19
> **Many thanks for the review, please find below our response**
>
> "In this paper, the authors present how to integrate replay buffer and on-policy trust region policy optimization (TRPO) by generalizing Q/V/advantage function and then empirically show the proposed method outperforms TRPO/DDPG."
>
> “The generalization of advantage function is quite interesting and is well written. One minor issue is that d^{\pi_n} (s) is confusing since it appears after ds. “
> Many thanks, we’ve made the necessary corrections.
>
> “The theory in Section 3.1 makes sense. However, due to the limitation in Theorem 1 that $\theta$ is the joint parameters, applying Theorem 1 can be difficult. In Eq (25), what is the $\theta$ here? And what does $\nabla_\theta \pi_n$ mean? Does $\pi_n$ uses $\theta$ for computation? One of the problems of using replay buffers in on-policy algorithms is that the stationary distribution of states changes as policy changes, and at least the writing doesn't make it clear on how to solve distribution mismatching issue. Further explanation on Eq (25) might help. If the distributions of states are assumed to match, then the joint distribution of states and actions may mismatch so additional importance sampling might help, as suggested in [1] Eq (3). “
>
> We’ve made corrections to the explanation of the equation (25). As we say in the text below this equation,
> "To practically implement this gradient, we substitute the parameters $\theta^\pi$, derived from the latest policy for the replay buffer, instead of joint $\theta$ parameters assuming that the parameters would not deviate far from each other due to the trust region restrictions; it is still possible to calculate the estimation of $\tilde{A}^{\pi_n} (s^n_t, a^n_t)$ for each policy using Equation (\ref{AdvantageEstimator}) as these policies are observed. "
>
> “Another problem is on the barrier function. In Eq (26), if we only evaluate $\rho_b(\theta)$ (or its gradient w.r.t. $\theta$) at the point $\theta_old$, it doesn't differ with or without the barrier function. So in order to show the barrier function helps, we must evaluate $\rho_b(\theta)$ (or its gradient) at a point $\theta \neq \theta_old$. As far as I know, the underlying optimizer, K-FAC, just evaluates the objective's (i.e., $\rho_b$) gradients at $\theta_old$. Both Conjugate Gradient (CG), which TRPO uses, and K-FAC are trying to solve $F^{-1} g$ where $g$ is the gradient of the objective at the current point. “
> It is also our understanding that the gradient in K-FAC is estimated at the current point only. That is why, when we use the K-FAC optimiser in practice for policy update, we do not just use it for one iteration but for several (N_ITER_PL_UPDATE = 10) iterations.
>
> The experiments show significant improvement over TRPO/DDPG. However, some experiments are also expected.
> "1. How is the proposed algorithm compared to PPO or Trust PCL? "
> We’ve added the experimental results to compare the improvement over PPO (see Figure 1).
>
> "2. How does the barrier function help? More importantly, what's the comparison of the barrier function to [1] Eq (5)? "
> The point behind the barrier function is to remove heuristic adjustments of the step parameter in optimisation. In TRPO and ACKTR, the step size is changed heuristically in order to make the solution fit the constraint; with the barrier function, we can perform the optimisation just with a fixed step size 0.01  applied to all tasks (to make sure we meet the repeatability requirements, we state these parameters in Appendix A). For PPO, the heuristic update rules for penalisation coefficient look like: if constraint < constraint_target / 1.5, step_size <- step_size/2; if constraint > constraint_target * 1.5, step_size <- step_size * 2 (it is our understanding that this procedure is based upon the heuristic rather than has theoretical background). We do not use it and use fixed barrier function constraint 100. Our point is that we maintain the performance without these heuristics, just with the hyperparameters, universal for all tasks.
>
> “The proposed algorithm seems more like a variant of ACKTR instead of TRPO since line search is missing in the proposed algorithm and the underlying optimizer is K-FAC instead of CG.”
> We agree with the statement that the methodology is closer to ACKTR while it differs significantly in various aspects (it includes barrier function optimisation, diagonal matrix adjustment, and, which is the main point of the paper, the replay buffer for TRPO); as it is outlined in Algorithm 1, we use the K-FAC  optimiser only for the policy gradient optimisation, while for the value function we use the Adam method, as we have seen no evidence of advantage of using K-FAC for the value function.

---

### Official Review · AnonReviewer3 · 2018-11-01
**Interessting work but some open questions remain**

**Rating:** 6
**Confidence:** 3

**Review:**

The authors introduce a off-policy method for TRPO by suggesting to use replay buffers to store trajectories and sample from them during training. To do this they extend the definition of the Q function to multiple policies where the Q_pi bar is then the expectation over the several policies. They propose the same for the value function and consequently the advantage function.
In my opinion this is some interesting work, but there are some details that are not clear to me, so i have several questions.

1. Why is it necessary to define these generalized notions of the Q, Value and Advantage functions? You motivate this by the fact the samples stored in the replay buffer will be generated by different policies, i.e. by differently parametrized policies at a certain time step. But this also holds almost all algorithms using replay buffers. Could you plese explain this part further?

2. In eq. (26) you introduce the parameter alpha as a sort of Lagrange multiplier to turn the unconstrained optimization problem defined by TRPO into a constrained one. This is was also proposed early by Schulman et al. in Proximal Policy Optimization. Yet, it is not cited or referenced. In the discussion of the experimental results go further into this. Please explain this part in more detail.

3. Another point of your work is the learnable diagonal covariance matrix. How can you be sure that the improvements you show are due to the replay buffers and not due to learning these? Or learning covariance in combination with the penalty term alpha?

4. Can you provide comparative results for PPO? PPO outperforms DDP and TRPO on most tasks so it would be interessting to see

5. How many trajectory samples do you store in the replay buffers? Can you provide results where you use your method but without any replay buffers, i.e. by using the last batch of data points?

Minor Suggestions:
- The references for the figures in the Experiments part are off. In fig. 1 you cite Todorov et al. for Mujoco but not TRPO and ACKTR, the same in fig. 2. Then in fig. 3 you cite DDPG also with Todorov et al.
- Some parts of the text is a bit unorganized. In section 2.1 you introduce AC algorithms and on the next page you give the definitions for all components but you don't say anything about how the interact. Also, the definition of the expected return was not "invented" by Schulman et al, and neither were Advantages, Q-, and Value functions. Maybe add a second or third reference.

---

> ### Author Response · Authors · 2018-11-19
> **Many thanks for the review, please find the the response and the revised version of the paper**
>
> 1.In many of the on-policy methods, such as TRPO, the distribution is assumed to be stationary (Sutton, 2000). This is in contrast to the methods based on deterministic policy gradient (Silver et al, Deterministic Policy Gradient Algorithms, ICML, 2014). We’ve amended the text in order to make this motivation clearer (see Introduction and section 3.1 for details). Using the generalised Q- and advantage functions formulation, we avoid the restriction of policy stationarity, imposed in policy gradient theorem which is often used for on-policy learning (Sutton et al, 2000).  We’ve amended the description around Equations (8)-(12) to reflect this.
>
> 2. While the optimisation methods themselves are not in a focus of the paper, we totally agree that we need to cite Schulman et al, as they apply a similar technique in their method (the main difference is that we have the \alpha parameter fixed whilst Schulman et al use adaptive parameters). We’ve modified the description accordingly.  Our proposal has happened to come independently from Schulman et al, although we believe it is because this technique seems to be the natural solution for constrained optimisation. The reason behind using this technique is to move away from heuristically defined dynamic estimation of the step size to make the problem fit the constraints, such as the one featured in ACKTR, and switch to one universal hyperparameter. In Proximal Policy Optimisation (see Section 4 of Schulman et al, 2017), the (heuristic) adaptive coefficient is still used.
> Our response to the reviewer 1 states that
> "The point behind the barrier function is to remove heuristic adjustments of the step parameter in optimisation. In TRPO and ACKTR, the step size is changed heuristically in order to make the solution fit the constraint; with the barrier function, we can perform the optimisation just with a fixed step size 0.01  applied to all tasks (to make sure we meet the repeatability requirements, we state these parameters in Appendix A). For PPO, the heuristic update rules for penalisation coefficient look like: if constraint < constraint_target / 1.5, step_size <- step_size/2; if constraint > constraint_target * 1.5, step_size <- step_size * 2 (it is our understanding that this procedure is based upon the heuristic rather than has theoretical background). We do not use it and use fixed barrier function constraint 100. Our point is that we maintain the performance without these heuristics, just with the hyperparameters, universal for all tasks. "
>
> 3. Indeed, the improvement is reasoned by different aspects of the method; we separate different aspects of the method, providing the comparison of the entire method in Figure 1, and elaborating on the influence of the replay buffers separately in Figure 2. In Figure 2, the buffer size 1 means that there is effectively no replay buffer, so the optimisation is carried out over the most recent policy only (i.e. without taking an advantage of the previous policies).
>
> 4. In the new revision of the paper, we provide PPO results (see Figure 1 for details).
>
> 5. We provide the results for our method without any replay buffers in Figure 2, and we’ve changed its description accordingly. Considering the trajectory samples, this number is not fixed as we put a cap on the number of overall samples per policy (no less than 1000 samples and until the terminal state, see appendix A). It typically corresponds to several(2-10) trajectory samples per policy.When we state that the size of the replay buffer is 3, it would mean that the samples are taken from 3 policies. It would mean no less than 3000 actions in the replay buffer. We've also added a statement the description of experimental section to reflect that the replay buffer size is the number of stored policies outputs rather than the number of samples in those policies which may vary.
>
> Minor Suggestions:
> "- The references for the figures in the Experiments part are off. In fig. 1 you cite Todorov et al. for Mujoco but not TRPO and ACKTR, the same in fig. 2. Then in fig. 3 you cite DDPG also with Todorov et al."
> We’ve fixed these omissions, many thanks for pointing at them.
> "- Some parts of the text is a bit unorganized. In section 2.1 you introduce AC algorithms and on the next page you give the definitions for all components but you don't say anything about how the interact. Also, the definition of the expected return was not "invented" by Schulman et al, and neither were Advantages, Q-, and Value functions. Maybe add a second or third reference.  "
> Totally agree, we’ve made the appropriate change in the descriptions; also we changed section 3.1.

---

### Official Review · AnonReviewer2 · 2018-11-02
**Seems a trivial extension of TRPO**

**Rating:** 7
**Confidence:** 5

**Review:**

The paper tries to bring together the replay buffer and on-policy method. However, the reviewer found major flaws in such a method.

- Such replay buffers are used for storing simulations from several policies at the same time, which are then utilised in the method, built upon generalised value and advantage functions, accommodating data from these policies.

If the experience the policy is learning from is not generated by the same policy, that is off-policy learning.

In the experiment part, the replay buffer size is often very tiny, e.g., 3 or 5. The reviewer believes there may be something wrong in the experiment setting. Or if the reviewer understood it incorrectly, please clarify the reason behind such a tiny replay buffer.

---

> ### Author Response · Authors · 2018-11-19
> **Thank you for the comment, please find detailed explanations below**
>
> “If the experience the policy is learning from is not generated by the same policy, that is off-policy learning. “
>
> The paper strongly relies on on-policy data collected with actual policies; the archive of previous policies is only to augment the training data, therefore it combines advantages of on-policy learning with also using (off-policy) replay buffer data for the previous policies. We've made modifications throughout the text to reflect this statement. As it is said in (Sutton&Barto, Reinforcement Learning: An Introduction, 2018):
> “All learning control methods face a dilemma: They seek to learn action values conditional on subsequent optimal behavior, but they need to behave non-optimally in order to explore all actions (to find the optimal actions). How can they learn about the optimal policy while behaving according to an exploratory policy? The on-policy approach in the preceding section is actually a compromise—it learns action values not for the optimal policy, but for a near-optimal policy that still explores. A more straightforward approach is to use two policies, one that is learned about and that becomes the optimal policy, and one that is more exploratory and is used to generate behavior. The policy being learned about is called the target policy, and the policy used to generate behavior is called the behavior policy. In this case we say that learning is from data “off” the target policy, and the overall process is termed off-policy learning. “
> The question we  pose in the paper is: maintaining the ability of on-policy learning to use the actual policy, would it be possible to augment it with the idea of replay buffers? In the proposed method, we do not use the separate behaviour policy to generate behaviour; instead we maintain the buffer of previous policies’ outcomes.   In the introduction, we have also put some emphasis on recent efforts to put together on-policy and off-policy learning (Nachum et al, TRUST-PCL: an off-policy trust region method for continuous control , ICLR, 2018; Gu et al, Interpolated Policy Gradient: Merging On-Policy and Off-Policy Gradient Estimation for Deep Reinforcement Learning, NIPS 2017).
>
> “In the experiment part, the replay buffer size is often very tiny, e.g., 3 or 5. The reviewer believes there may be something wrong in the experiment setting. Or if the reviewer understood it incorrectly, please clarify the reason behind such a tiny replay buffer.”
>
> The size of the replay buffer, in our case, is measured in terms of the number of policies, not actions (i.e. it would mean that given that we collect at least 1000 actions  from a single policy as outlined in appendix A, buffer size 3 would mean samples from three policies, i.e. no less than 3000 samples in the buffer). We have made sure that this is evident the text of the paper (see the revised experimental section). The motivation behind the notion of the replay buffer in this paper is different from the one for the state-of-the art off-policy replay buffers (like DDPG). Rather than leveraging upon a multitude of previous policies, we use replay buffers to better approximate the vicinity of the current policy within the trust region policy optimisation approach and therefore save the simulation steps numbers.
>
> We believe that the novelty of our approach is that using the formulation of the generalised Q- and advantage functions (see Equations (8)-(12)) we avoid the restriction of policy stationarity, imposed in policy gradient theorem which is often used for on-policy learning (Sutton et al, 2000).  We’ve amended the description around Equations (8)-(12) to reflect this. Theorem 2 highlights the reason behind a small replay buffer: as it is said in the comments below the theorem, “given the continuous value function approximator it reveals that the closer are the policies, within a few trust regions radii, the smaller will be the bias” in advantage function approximation. Deterministic Policy gradient (Silver et al, Deterministic Policy Gradient Algorithms, ICML, 2014) does not have this assumption of policy stationarity, and it allows it to maintain far larger replay buffers.
>
>  We’ve made modifications in the paper to make the difference more clear.

---

### Author Response · Authors · 2018-11-19
**Summary of the amendments**

The authors would like to thank the reviewers, whose contributions, hopefully, helped to improve the quality of the paper.

The minor comments of reviewers 1 and 3 have been addressed in the corrections; necessary changes on the major comments from all reviewers are listed in the comments below.

Below is the summary of the major changes in the second revision:
- in the abstract and introduction, the emphasis is put on the motivation behind using data from the previous policies alongside with the off-policy data
- in the introduction, new references on the relevant methods, benefitting from both on-policy and off-policy data (Trust-PCL, IPG), are given, as well as the reference on PPO which is also relevant to the proposed method
- the comment, addressing practical implementation of gradient in eq. (25) and practical application of the Theorem 1, has been added after eq. (25)
- the experimental section now include new results for PPO
- the clarifications on the meaning of replay buffer size in Figures 1-3 have been added to the experimental section

---

### Meta-Review · Area_Chair1 · 2018-12-12
**Issues with the proposed gradient**

**Confidence:** 4
**Recommendation:** Reject

**Metareview:**

The reviewers raise an important issue about the parameters in the proposed gradient in Theorem 1. There could be different parameters for each policy in the gradient (though some parameter sharing could be possible), and computing this gradient would be prohibitive. The solution is to just use the most recent parameters, but then the gradients become off-policy again without motivation for why this is acceptable. This approximation needs to be better justified.

As an additional point, there are other off-policy policy gradient methods, than just DPG. The authors could consider comparing to these strategies (which can use replay buffers) and explain why the proposed strategy provides benefits beyond these. What is inadequate about these methods? Further motivation is needed for the proposed strategy. This is additionally true because the proposed strategy requires entire sampled trajectories for a fixed policy (to make the policy gradient sound, with weighting dpi_n(s)), whereas DPG and other off-policy AC methods do not need that.